# Mediation of a Mutualistic Conflict for Pollination via Fig Phenology and Odor Recognition between *Ficus* and Fig Wasp

**DOI:** 10.3390/plants11192603

**Published:** 2022-10-03

**Authors:** Wen-Hsuan Chen, Anthony Bain, Sheng-Yang Wang, Yi-Chiao Ho, Hsy-Yu Tzeng

**Affiliations:** 1Department of Forestry, National Chung Hsing University, Taichung 40227, Taiwan; 2Chiayi Forest District Office, Forestry Bureau, Council of Agriculture, Executive Yuan, Chiayi City 60000, Taiwan; 3Department of Biological Sciences, National Sun Yat-sen University, Kaohsiung 80424, Taiwan; 4International PhD Program for Science, National Sun Yat-sen University, Kaohsiung 80424, Taiwan; 5Academy of Circular Economy, National Chung Hsing University, Taichung 40227, Taiwan; 6Hsinchu Forest District Office, Forestry Bureau, Council of Agriculture, Executive Yuan, Chiayi City 30191, Taiwan

**Keywords:** dioecy, *Ficus*, fig phenology, volatile compounds, pollination mutualism

## Abstract

The vegetative and reproductive growth of plants provide the basic tempo for an ecosystem, and when species are interdependent, phenology becomes crucial to regulating the quantity and quality of the interactions. In plant–insect interactions, the plants signal the beginning of their reproductive period with visual and chemical cues; however, in the case of *Ficus* mutualism, the cues are strictly chemical. The volatile organic compounds emitted by a fig species are a unique, specific blend that provides a signal to mutualistic wasps that the figs are receptive for pollination. In this study, we studied both the phenological pattern of *Ficus septica* in Central Taiwan and its emissions of volatile compounds at receptivity. This dioecious fig species displays a pattern of continuous vegetative and reproductive production all through the year with a decrease in winter. In parallel, the odor blends emitted by male and female trees are similar but with seasonal variations; these are minimal during winter and increase with the size of the wasp population during the favorable season. In addition, the pollinating females cannot distinguish between the male and female summer odor blends. The link between odor similarity, pollinators and intersexual conflict is discussed.

## 1. Introduction

Phenology is the study of biological events from which many ecological processes derive [1]. Plant flowering is a major phenological event in ecosystems because it drives the abundance and diversity of pollinating species [2,3]. The relationship between flowering plants and their pollinators is often mutualistic through pollination, and the landscape can be considered as a network of mutualisms where many species are linked with others via a series of interactions [4,5]. Mutualisms are common and take many forms, ranging from facultative to obligate, symbiotic or not [6]. *Ficus* (Moraceae) is one of the most speciose tree genera with about 750 species worldwide characterized by the enclosed inflorescence known as fig or syconium [7]. Figs are famously known for their obligate nursery pollination mutualism with pollinating fig wasps (Hymenoptera: Agaonidae) [8]. Fig wasps are minute insects that live for a few hours up to a few days [9]; they pollinate specific figs by entering the enclosed fig via the only entrance, known as the ostiole [10,11]. The specificity of each *Ficus*–fig wasp pair is shaped by the morphology of the ostiole [12] but it is mostly driven by the recognition of the volatile organic compounds (VOCs) emitted by each fig species [13,14,15]. Even though many of the emitted VOCs are the same, the composition of the odor bouquet is different [16] with several specific compounds being the key to the recognition [17].

About half of the *Ficus* species are functionally dioecious [7] and the specificity of the dioecy in the fig wasp mutualism is that the mutualistic interaction only occurs in male fig trees whereas the female fig trees exploit the pollinating wasps without rewarding them. Indeed, when a pollinator enters a female fig, it pollinates but cannot lay eggs due to the extended length of the ovule in female figs [18]. The wasp dies without offspring but the female trees produce seeds. The conflict between wasps and fig trees and the necessity for the trees to conceal their sexual identity are an advantage to both male and female trees [19]; when a pollinating wasp pollinates a female fig, the two parents of the future seeds have succeeded in reproducing. Thus, male and female figs of synchronous species emit the same odor [16], making them indistinguishable to the wasps; if it were not so, the wasps are likely to choose male trees, leading to the species extinction [18]. In this study, we aimed to explore the links between the reproductive phenology of *Ficus septica* and the intraspecific mimicry of the fig smell. As *F. septica* is a very common fig species in Taiwan, with a very wide distribution, we want to understand how it handles the conflict of being very successful in the environments it has colonized. The scientific questions behind this statement are simple: (1) from the point-of-view of reproductive phenology, how do *F. septica* trees maintain their population of short-lived fig wasps, and (2) how do they avoid losing too many pollinators to the female figs? We hypothesized that pollinating fig wasps are not able to distinguish male figs from female figs and thus do not avoid female figs.

## 2. Results

### 2.1. Phenological Patterns

Figs were observed on fig trees from November 2008 to December 2013 (Figure 1). All the male trees bore figs during 242 surveys (94.5%) while the value for the female trees was slightly lower: 210 surveys (82.0%) with all the female trees bearing figs. Unpollinated figs were observed in 91.8% of the surveys on male trees and 81.4% of the surveys on the female trees during our observation period.

The general phenological pattern for both sexes demonstrates the continuous production of figs with several peaks from the end of the winter to the end of the following autumn (Figure 1). Similarly, new leaves were produced all year long with a peak at the beginning of the year and older leaves turned yellow all year long with a clear peak around November each year. Moreover, the number of new leaves and yellow leaves did not differ between years, but the number of male and female unpollinated figs changed (Kruskal–Wallis test: *p* < 0.001 for both sexes). The vegetative phenological characteristics did not display differences between the two sexes and were positively correlated together (Table 1). Additionally, the reproductive and non-reproductive phenologies of the male trees were strongly correlated, whereas for the female trees, only the A phase figs displayed a significant correlation with the total number of figs and the number of young leaves (Table 2).

Meteorological factors have numerous effects on the phenological features (Table 3); however, the correlation coefficients were weak with a maximum of 0.183. Two main categories of phenological features can be analyzed: reproductive (figs) phenology and vegetative (leaves) phenology. Of the 60 tested correlations, 31 (51.7%) were significant for the vegetative phenology against only 14 (23.3%) for the reproductive phenology. The significant correlations of the vegetative phenology were in relation to female growth (60% (18) of the time). Additionally, 64.3% (9) of the significant correlations for the reproduction phenology were for female trees. Moreover, overall, almost half of these positive correlations were with values of relative humidity (15 correlations, Table 3).

### 2.2. Volatile Composition of Receptive Figs and Seasonal Changes

A total of 31 compounds were detected in the headspace collections from the two genders at the receptive phase and 28 (90.3%) were identified and categorized into three classes (see Appendix A, Appendix A for the complete list of compounds). These included benzenoid (one compound, 3.6%), monoterpenoids (15 compounds, 53.6%), and sesquiterpenoids (12 compounds, 42.8%). The male samples were more diverse than the female samples, which comprised 31 and 19 compounds, respectively. The most abundant compounds in the male and female samples were eucalyptol (32.2% of the relative amount in males and 28.8% in females), linalool (21.0% and 11.1%), sabinene (5.61% and 16.3%), and trans-β-ocimene (7.76% and 12.0%), which together represented 66.7% of the relative amount in males and 68.1% in females (Appendix A, Appendix A).

The PCA analysis did not display a very clear pattern with the exception of one central group of samples from 2010 (Figure 2). The remaining points were scattered around the 2010 cluster. The PCA’s two axes explained 69.6% of the variance, which was mainly due to the amount of eucalyptol and linalool (Appendix A, Appendix A for details).

The similarity of the volatile compounds in receptive figs between the genders ranged from 46.4% to 84.7% (Figure 3). The August samples had similar percentages but the highest value was from June 2011 (Figure 3). Moreover, the two lowest values were from April 2011 and September 2010.

### 2.3. Y-Tube Olfactometer Bioassay

The pollinating wasps *Ceratosolen* sp.1 were significantly attracted by both female and male receptive fig odors but no tests revealed any significant difference between the male and female odors when tested together (Figure 4). 

## 3. Discussion

Figs were always present on the studied trees. Individuals from both sexes had peaks in production several times a year but mostly during the first months of the year. The pattern was similar for the production of new leaves but more punctual than for the figs. The correlations with the meteorological factors were very weak when they existed. On the other hand, the positive correlation between the male A phase figs and leaf growth is an important indicator of the physiological pressure of the fig production. This correlation has not been observed often in other fig species and when it was found, the other study methods did not take autocorrelation into account ([20] but see [21,22]). Thus, the fig mass increases rapidly after pollination [21], the trees need a significant intake of carbon and new leaves are more efficient in photosynthesis [23,24]. So, fig production may regulate the vegetative phenology in order to sustain the significant nutrient demand of the fig crop.

The fruiting phenological pattern in this study is similar to other *F. septica* phenology studies in Taiwan [23,24] and to species from the same taxon (subgenus Sycomorus) in Asia: *F. hispida* [25,26], *F. squamosa* in Thailand [20], and *F. fistulosa* in Singapore [27]. Tropical fig species from different subgenera also present similar phenological patterns such as *F. fulva* (subgenus *Ficus*) on the island of Borneo [21] and *F. pedunculosa* var. *mearnsii* [28,29]. Studies that show continuous fig production within a population are numerous and widespread enough to define fig production as continuous at a population level. An efficient and widespread strategy is used by *Ficus* species to maintain the population of their mutualistic pollinator as well as feeding seed dispersers. Nevertheless, studies like this one have shown the phenological differences between male and female trees, that is, female individuals produce less figs and have fewer production peaks ([23,24,30,31], this study). Thus, female trees have not been selected to produce frequent and asynchronous crops like male trees, which are required to maintain the population of fig wasps [10,24].

Moreover, in dioecious fig species, there is conflict between male and female trees because the female figs produce neither pollen nor wasps; they are a pollinator sink [18,19]. Indeed, when a female pollinator enters a female fig, the wasp will not produce offspring and will die inside the fig. Patel et al. [18] have asked why these pollinators commit suicide by entering female figs. They advance two main hypotheses, the “no available choice” and the “no preference” hypotheses. The former describes the situation where only one sex has available figs, leading to the absence of choice for the fig wasps, as has been described for *F. erecta* var. *beechayana* [32,33] and *F. carica* [34]. This hypothesis can be excluded for *F. septica* as these trees produce figs frequently without any recognizable pattern. On the other hand, the “no preference” hypothesis states that the pollinating wasp cannot differentiate between the sex of the figs [18]. The hypothesis has been validated for *F. carica* (therefore, *F. carica* uses both strategies to maintain its pollinator population). When trees of both sexes are producing figs synchronously in summer, their odors are more similar than for other crops ([35,36] but see [37]). This trend has also been observed in many tropical dioecious species from three *Ficus* subgenera [16]; when trees of both sexes are producing figs synchronously, the fig odors are statistically undistinguishable. Nevertheless, the intraspecific variation in the odor blend was stated to have no data in a 2010 review [38] but the male *F. carica* trees have so much variability in their smell that it is statistically different [35]. In addition, more recently, a few studies have examined the geographical variation in fig odor and have shown that the compounds emitted by figs are not uniform over the species distribution [17,39].

The conflict between the pollinating wasps and the female fig trees is important because if the wasps successfully identify the fig sex, they will avoid female figs, and eventually, the species will become extinct. This statement leads to a question about the origin of dioecy in *Ficus*: how variable was the fig smell in the first dioecious fig species? Fig odor blends at receptivity are species specific [38,39] but very little is known about monoecious species (but [17] about *F. racemosa*). Additionally, *F. racemosa* is quite special as it is from a group of dioecious species that reverted to monoecy [7].

Nevertheless, we can assume that the first dioecious *Ficus* species were very similar in terms of their odor blend as they were very successful and are now speciose (half of the genus *Ficus*). Secondly, we can also assume that the intraspecific variation is minimal because when dioecious species speciate, they need to have a uniform fig smell in order to successfully speciate apart. Thirdly, we can assume that the intersexual sex mimicry is “reset” at each new speciation event. Therefore, how to explain the relaxation of the interspecific mimicry in *F. septica*?

As described above, most of the studies on fig smell have been conducted on Mediterranean and temperate *F. carica* trees. Male *F. carica* are required to have some pollinated figs for their pollinators to overwinter, thus the last pollination and fig receptivity of the year are critical for *F. carica*. This is not the case in our study species as *F. septica*, can produce figs through winter (Figure 5), even if the wasp population and fig production in Taiwan are decreased in winter [40]. Like most places in the northern hemisphere, winter is the harshest season in Taiwan, and *Ficus* phenology shows that the crops are less frequent and need a longer development period [24,32,41,42]. Therefore, it may not be the optimal period for growing fruits and seeds. Moreover, *F. septica* is distributed mostly in intra-tropical areas and Taiwan is at the northernmost border of its distribution [7].

In these conditions, the intersexual mimicry of fig smell, outside of the main wasp season, may have been counter-selected in a place like Taiwan. When female trees have a different fig smell, they may attract fewer pollinators, thus less figs will grow during the coldest periods. On the other hand, in the summer and spring, the fig smells are similar to the male fig smells to prevent the pollinators from knowing the sex difference. This phenomenon may be peculiar to tropical species living in a highly seasonal and subtropical environment. Nevertheless, it may be a selected feature as it mitigates the winter investment in fruits and diminishes the loss of pollinators during the seasonal low in their population.

Fig phenology and the emission of volatile compounds are closely linked [16] and our study shows that the link is also influenced by the local environment. Furthermore, the conflict between female figs and pollinating wasps has attracted the interest of many researchers because of the ubiquity of the exploitation by female fig trees even though the selective pressure on the survival of the pollinating wasps is so extreme that it even shapes the conditions of their exploitation. Further studies that widen the sampling of *F. septica* across its distribution may show that the intersexual mimicry is an adaptive characteristic to temperate environments with equatorial populations showing constant intersexual mimicry.

## 4. Materials and Methods

### 4.1. Species and Site

*Ficus septica* Burm. f. is one of the common dioecious figs in Taiwan, belonging to the subgenus *Sycomorus* section *Sycocarpus* [7,43]. It is a small- to medium-sized free standing tree, approximately 1.5 to 6 m high. In Taiwan, *F. septica* is pollinated by three *Ceratosolen* wasp species [44] but only *Ceratosolen* sp.1 has been observed in the study area.

Fig development is divided into five distinct phases [45]: (1) A phase (pre floral); (2) B phase (receptive); (3) C phase (inter floral); (4) D phase (emergence); and (5) E phase (ripening). Fig development is bound to the life cycle of the pollinating fig wasp. Indeed, the female wasps enter and pollinate the fig during the receptive period of the figs’ development, the B phase; then, the maturation of the wasp larvae and the fig seeds occur during the C phase. The wasps are enclosed before the figs ripen during the D phase and the female pollinating wasps leave their native figs after mating. Finally, during the E phase, the figs ripen. Moreover, for dioecious fig species, the two final phases are segregated between male (D phase) and female (E phase) trees (Ho et al. 2011).

The studied trees were located at Sue-Te Park, Taichung, Central Taiwan (N 24°7′3.67″, E 120°39′14.17″; alt. 52 m). The area is classified as having a humid and warm temperate climate with a hot summer [46], the yearly average temperature is 23.1 °C and the average annual precipitation is 1676 mm (1961–2011) with clear seasons: about 79% of the rainfall occurs between May and October (Central Weather Bureau of Taiwan, Taipei, Taiwan, www.cwb.gov.tw/eng/, accessed on 1 September 2022).

### 4.2. Field Censuses

The fig and leaf phenology of 14 *F. septica* trees (seven males and seven females) were monitored weekly (average: 7.4 days) from November 2007 to January 2013 (256 surveys). On each individual, eight to 12 branches (30 cm long) were marked and the fig number and their developmental phase were noted at each survey (Ho et al., 2011). During the survey period, four female trees died because of artificial cutting or insect pests, and no additional trees were available in the area to replace them.

### 4.3. Collection of Volatile Compounds

The period between the appearance of fig buds and fig receptivity (B phase) can be extremely short (one to a few days), and in the studied area, the B phase often lasts only few hours before the figs are pollinated. Thus, due to the difficulty of finding unpollinated B phase figs, samples from different individuals were pooled together for the odor collection, but figs of different sexes were kept apart (as per the methods used in [14]). A male odor pool (seven individuals) and a female odor pool (three individuals) were extracted at each of the sampling surveys: August to October 2010 and, January to August 2011. To ensure that figs were not pollinated, unpollinated (but not yet receptive) figs were covered with mesh bags in order to exclude pollinators. When these figs reached the receptive phase, the volatile compounds were collected. Odor collection was typically performed outside on a sunny morning between 08:00 and 12:00 during the phenology surveys. Branches carrying B phase figs were cut from the tree, leaves and figs in other phases were removed, and the branches were directly sealed inside a polyethylene (PE) bag. Control air samples were also taken simultaneously in the field.

For volatile compound collection, we used solid-phase micro-extraction (75 μm carboxen-PDMS, Supelco^®®^, Sigma-Aldrich Co. LLC., Burlington, MA, USA), and the scent of receptive figs was collected using the dynamic headspace technique. The collection of fig scents and air control samples began 30 min after the branches were cut and put into the PE bags. The samples were analyzed by gas chromatograph-spectrometry (GC-MS), using a Varian GC CP 3800 chromatograph coupled with a 1200 mass spectrometry (Agilent Technologies Inc., Santa Clara, CA, USA). For both the gas chromatograph and spectrometer, a DB-5 column was used (30 m × 0.25 mm × 0.25 μm; Agilent Technologies Inc., USA). The instrumentation and temperature programs were as follows: electronic flow control was used to maintain a constant helium carrier gas flow of 1.0 mL/min, the GC oven temperature was held at 40 °C for 1 min, increased by 4 °C/min to 260 °C and maintained at that temperature for 4 min, the MS interface was 280 °C, and the mass spectra were taken at 70 eV (in EI mode) with a scanning speed of one per scan from m/z 45–425. The GC-MS data were processed using a Varian Workstation software package. Component identification was conducted by matching the mass spectra with the Wiley/NBS Registry and retention index (relative to n-alkanes), and confirmed by comparing the retention index and mass spectra with published data (http://webbook.nist.gov/chemistry/, accessed on 1 September 2022 ) or authentic standards when available.

### 4.4. Y-Tube Olfactometer Bioassay

The Y-tube olfactometer (stem: 8 cm; arms: 9 cm; at an angle of 55° angle; diameter: 1.5 cm) was used to test the responses of female pollinators *Ceratosolen* sp.1 to fresh fig odors during three periods: August 2010, May 2011 and August 2015. Each arm was connected to a Nalophan^®®^ bag containing an odor source. Air was drawn through a Teflon tube by an air pump and passed through a charcoal filter. The air stream was regulated to a flow rate of 200 mL/min with two flowmeters and was split via a Y-hose junction into two equal air streams. To avoid visual distractions for pollinators, the olfactometer was placed in the center of a box with one 5 W cool white fluorescent tube placed above the arms of the Y-tube. The box was kept dark with a black curtain. The air temperature was monitored and maintained at 25–27 °C.

Each pollinating wasp was tested independently, allowing three minutes in the olfactometer. The pollinator choice was noted (left or right arm) when it crossed the decision line of the arm (1 cm from the Y junction) and stayed there for more than 1 min. Wasps that did not reach the decision line after 3 min were removed and recorded as having exhibited “no choice”. The olfactometer was rinsed with 95% ethanol and then dried by an air blower after each bioassay, and after every four tests, the treatment arm was switched to avoid any influence of unforeseen asymmetries in the setup.

### 4.5. Data Analysis

Time series data are often autocorrelated [47] and nonparametric, as was the case for all the phenological data collected during this study when we used the Breusch–Godfrey test on the data. Thus, generalized least squares (GLS) models with autocorrelated errors were used to test the dependence of the various phenological time series data and the effects of the meteorological factors. The tests were done using the lmtest and nlme packages [48,49] in R v3.2.4 [50] as well as all the other statistical tests on the phenological data.

The phenological and meteorological (rainfall, temperature, relative humidity and solar radiation) data were tested for correlations over 7, 15, 30 or 45 day periods prior to the survey day. Typhoons passing over Taiwan bring significant rainfall; thus, in order to avoid any statistical bias, the extreme precipitation days (days with >160 mm, a value never obtained outside typhoon events) were removed from the analysis (eight days were removed).

We identified compounds in the odor composition of the fig and integrated the peak of every substance to a relative proportion. The relative proportion of the different emitted compounds was determined for each sample. Principal component analysis (PCA) was used to analyze the percentage of relative content using PC-ORD v5.0 [51]. Then we compared the relative amounts of all compounds and estimated the percentage similarity of volatile compositions for the same-month sample among sexes (Whittaker and Fairbanks 1958). The monthly percentage of similarity (PS) formula for both gender samples is
$$\mathrm{PS}=1-0.5\sum_{i=1}^{X}[p_{M_{i}}-p_{F_{i}}]$$
where $p_{M_{i}}$ is the proportion of the compound i for males (M), X compounds are in the blend of male and female figs at this stage, and $p_{F_{i}}$ is for the female (F) samples.

The proportion of wasps that made a “choice” vs. “no choice” was compared between the control and fig odors were added to the olfactometer using a 2 × 2 contingency table. For all comparisons, the response of wasps was analyzed using the χ^2^ test with Yates correction. The statistical tests on the VOC data were done using SPSS 12.0 (SPSS Inc., Chicago, IL, USA).

## Figures and Tables

**Figure 1 plants-11-02603-f001:**
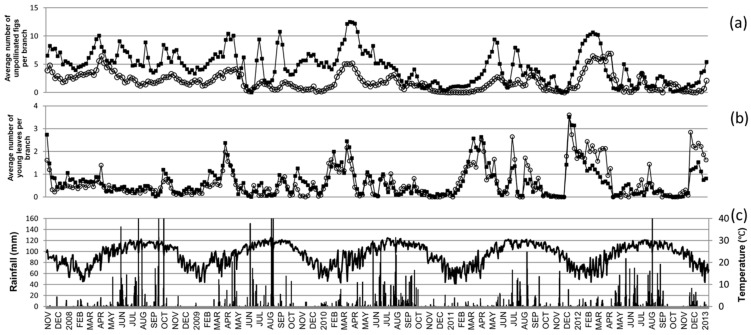
Reproductive and vegetative production in relation to the meteorological factors. Unpollinated (A-phase) figs (**a**); young leaves (**b**); and daily rainfall and average temperature during the survey period (**c**). The filled squares represent male production and the hollow circles represent female production (**a**,**b**). The black columns represent rainfall and the black line represents average daily temperature (**c**).

**Figure 2 plants-11-02603-f002:**
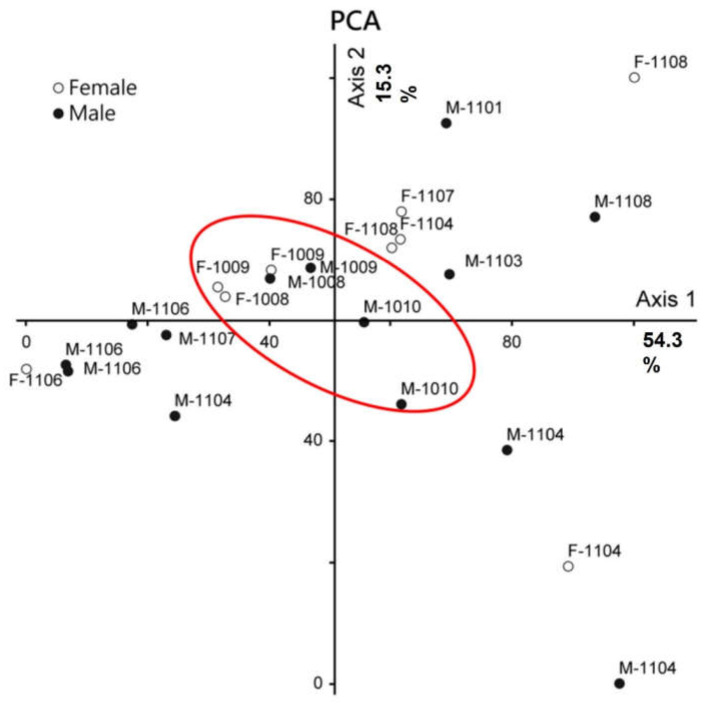
Principal component analysis (PCA) of volatile bouquets between receptive female (hollowed circle) and male (filled circle) figs; The fourth first digits of each triangle represent the date (yymm).

**Figure 3 plants-11-02603-f003:**
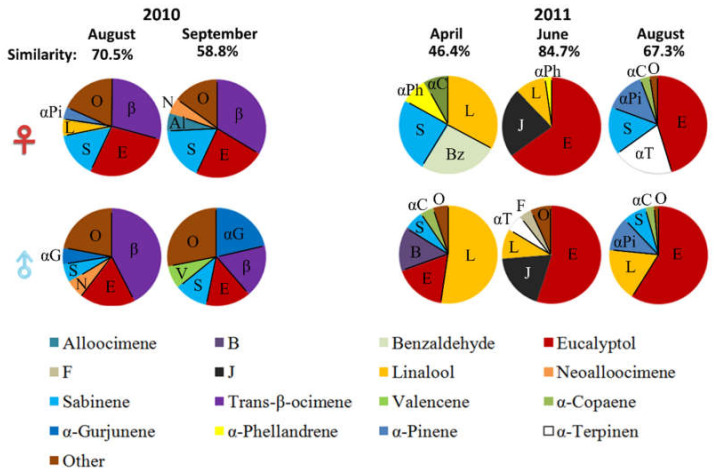
Seasonal changes in volatile compounds in receptive female and male figs of *Ficus septica* in central Taiwan.

**Figure 4 plants-11-02603-f004:**
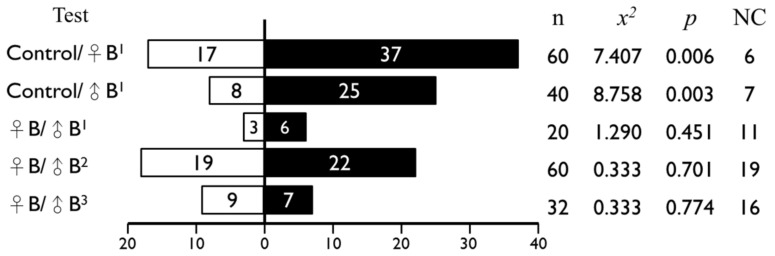
Number of female *Ceratosolen* responding to fig volatiles in Y-tube olfactometer. The superscript indicates the date of the tests: ^1^ 16.08.2010, ^2^ 26.05.2011 and ^3^ 20.08.2015. NC is “no choice”.

**Figure 5 plants-11-02603-f005:**
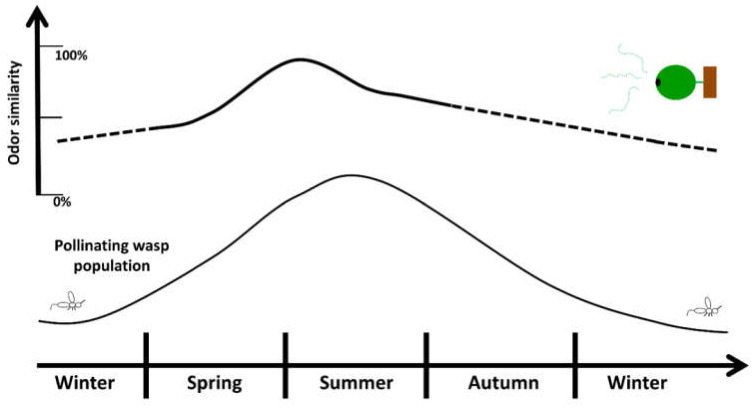
Wasp population and odor similarity trends over a year.

**Table 1 plants-11-02603-t001:** Generalized least squares (GLS) models with autocorrelated errors and the Wilcoxon signed rank tests between male and female phenological production by branch. V is the statistic associated with the Wilcoxon rank test.

	GLS		Wilcoxon Rank Test	
	Correlation Coefficient	*p*	V	*p*
Male A figs vs. Female A figs	0.666	**<0.001**	31,596	**<0.001**
Male figs vs. Female figs	0.352	**<0.01**	31,694	**<0.001**
Male young leaves vs. Female young leaves	0.555	**<0.001**	17,540	>0.05
Male yellow leaves vs. Female yellow leaves	0.301	**<0.001**	18,262	>0.05

**Table 2 plants-11-02603-t002:** Correlation coefficients of the generalized least (GLS) models with autocorrelated errors between the phenological characteristics of the male (below) and female (above) trees.

	A Figs	Figs	Young Leaves	Yellow Leaves
A figs	-	0.331 ***	0.209 *	NS
Figs	0.516 ***	-	NS	NS
Young leaves	1.099 ***	0.894 **	-	NS
Yellow leaves	0.903 *	1.288 **	NS	-

NS: Not Significant; * *p* < 0.05; ** *p* < 0.01; *** *p* < 0.001.

**Table 3 plants-11-02603-t003:** Correlation coefficients of the generalized least (GLS) models with autocorrelated errors between phenological characteristics and climatic factors for male (M) and female (F) production.

	A-Phase Figs		Total Figs		Young Leaves		Yellow Leaves	
	M	F	M	F	M	F	M	F
Rain								
15 days	−0.052 **	−0.018 *	−0.049 *	NS	NS	NS	−0.009 ***	−0.013 ***
30 days	0.074 *	NS	NS	NS	NS	NS	−0.008 *	−0.019 **
45 days	NS	NS	NS	NS	NS	NS	NS	−0.014 *
Temperature								
7 days	NS	NS	NS	0.068 *	NS	−0.037 **	NS	NS
15 days	NS	NS	NS	0.137 **	NS	−0.072 ***	NS	NS
30 days	NS	NS	NS	0.157 *	−0.047 ***	−0.076 ***	NS	NS
45 days	NS	NS	NS	0.163 *	−0.046 *	−0.070 **	NS	NS
Relative humidity								
7 days	NS	NS	NS	NS	NS	0.010 *	−0.006 **	−0.012 ***
15 days	NS	−0.023 *	NS	NS	NS	0.030 ***	−0.014 ***	−0.027 ***
30 days	0.098 *	0.047 *	NS	NS	0.058 *	0.080 ***	NS	−0.024 ***
45 days	NS	NS	NS	NS	0.045 *	0.060 **	NS	−0.019 *
Solar radiation								
7 days	NS	NS	NS	NS	NS	NS	NS	NS
15 days	0.092 *	0.046 *	0.119 *	NS	NS	−0.031 *	0.012 *	0.021 *
30 days	NS	NS	NS	NS	−0.071 ***	−0.108 ***	NS	NS
45 days	NS	NS	NS	0.183 *	−0.075 **	−0.122 ***	NS	NS

NS: Not Significant; * *p* < 0.05; ** *p* < 0.01; *** *p* < 0.001.

## Data Availability

Not applicable.

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
