# Peer review of "Mediation of a Mutualistic Conflict for Pollination via Fig Phenology and Odor Recognition between Ficus and Fig Wasp"

_plants, 2022, doi:10.3390/plants11192603_

Round 1

Reviewer 1 Report

This long-term study of a dioecious species of fig provides some interesting results - male and female syconia have very similar odor profiles the better to survive, as with different odors pollinating wasps may learn or be selected to avoid females that don't permit oviposition. I find it confusing that the authors use the word 'fig' to mean not only the type of tree and the fruit, but also the inflorescence itself. I feel that the word 'syconium' should be used for both male and female inflorescences, as only the female ones can develop fig fruits. Some of my comments about lack of clarity resulted from the methods section following the results. Maybe this can be remedied by more explanation of the terminology in the introduction, especially the use of the different phases (A, B, etc.) which could make the whole manuscript easier to understand. I have made numerous small comments on the pdf using the Comment Tool in Adobe, I hope they will be helpful in revision.

Author Response

The comments of reviewer 1 were edits posted directly in the pdf and they have been changed in the manuscript. The word "fig" was often suggested to be changed for the word "syconium" but the reviewer may not be aware of recent developments in the fig research: during the 10th fig symposium, that happened in July this year, a full presentation was made about the word syconium and it does not apply for most figs actually (and Ficus septica our stuudy species). For this reason, we have kept the word fig as it is.

Reviewer 2 Report

The manuscript presents an interesting topic that may have an impact on crop management.

The study was conducted with great care and attention.

The introduction with a good bibliography showed the state of the art and the reasons for this investigation.

The methodology was well prepared and the data analysis well conducted.

The results have been clearly stated and accompany the reader in understanding the study.

I just highlight a few things that should be reviewed:

- In table 1 indicate what V means.

- In line 126, the percentage of variance explained is expressed. However, it should be indicated how much percentage should be attributed to the two axes. It would be even better if these percentages were reported in the two axes of the graph in Figure 2.

- In figure 3 the percentages in the pie charts are indicated only by colors. Color blind people may not understand the information displayed. When using colors, it is always a good idea to try creating a grayscale pdf file. In this way you experience the most severe mode of color blindness. If the differences remain perceptible then that's okay, otherwise you have to find solutions. In this case, you can add numbers or letters identifying the volatile compounds and delineate the boundaries between the slices of the cake.

Author Response

- In table 1 indicate what V means.

[The meaning of V has been added to the table caption]

- In line 126, the percentage of variance explained is expressed. However, it should be indicated how much percentage should be attributed to the two axes. It would be even better if these percentages were reported in the two axes of the graph in Figure 2.

[The percentage has been added to the two axis on Figure 2.]

- In figure 3 the percentages in the pie charts are indicated only by colors. Color blind people may not understand the information displayed. When using colors, it is always a good idea to try creating a grayscale pdf file. In this way you experience the most severe mode of color blindness. If the differences remain perceptible then that's okay, otherwise you have to find solutions. In this case, you can add numbers or letters identifying the volatile compounds and delineate the boundaries between the slices of the cake.

[The figure 3 has been changed: lines have been drawn between colors and letters have been added within each color pie part.]

Reviewer 3 Report

The manuscript presents an original view of the male and female summer odor similarity, pollinators, and intersexuality in Ficus septica. The whole manuscript is well-organized, well-written, and interesting. However, some minor remarks) exist, i.e., no hypothesis; the sentence (L.74) is unclear; editorial errors. I think that more interesting for readers would be presented Table S1 in the main paper, not in suppl. mat. 

Author Response

The manuscript presents an original view of the male and female summer odor similarity, pollinators, and intersexuality in Ficus septica. The whole manuscript is well-organized, well-written, and interesting. However, some minor remarks) exist, i.e., no hypothesis; the sentence (L.74) is unclear; editorial errors. I think that more interesting for readers would be presented Table S1 in the main paper, not in suppl. mat. 

[A hypothesis part has been added at the end of the introduction. The sentence has been clarified and the document checked for editorial errors. Nevertheless, we think that the Table S1 will only benefit to a small proportion of readers if it was in the figures of the article whereas readers interested specifically in which components were emitted by the figs of F. septica can find it in the supplementary materials of this paper.]